# Teaching a Biblical Text among African Christian and Muslim Asylum-Seeker Children in Israel

**Dolly Eliyahu-Levi * and Michal Ganz-Meishar**

Education Faculty, The Academic College Levinsky-Wingate, Shoshana Persitz 15 st., Tel Aviv 6937808, Israel
* Correspondence: doly-levi@l-w.ac.il

**Abstract:** Educators in Israel face significant school diversity while struggling to adequately respond to the unique needs of diverse national and cultural communities and students from different socioeconomic backgrounds. Bible teachers in elementary school face tensions and conflicts between the religious concepts and beliefs of the parents and the children and the accepted concepts in Israeli Jewish society. This qualitative study was conducted among fifteen teachers working in elementary schools in the country's center where students from national, religious, social, and social–cultural populations attend, including children from families of asylum seekers. The findings revealed two central tensions: (1) emotional religious tension and (2) pedagogical tension. It was found that Bible teachers play the role of social–religious mediators in Israeli society. In the context of religious tension, teachers find themselves in situations of uncertainty, without the pedagogical skills to help them bridge the gaps and soften the strain. As a result, they are passive and remain silent. On the other hand, in the context of pedagogical tension, the teachers try to take the initiative, go beyond the boundaries of the familiar and known, and try to adapt classroom activities to the culture of the country of origin and the everyday social contexts.

**Keywords:** biblical teaching; African asylum seekers; pedagogical tension

## 1. Introduction

Israel is not an isolated island in the global migration process. It should provide an educational, sociocultural solution to integrating tens of thousands of African asylum seekers. A very significant number of these asylum seekers stay illegally, living in difficult conditions on the social margins while suffering from general, occupational, and family instability and experiencing a sense of alienation and social non-belonging (Kritzman-Amir 2015; Della-Pergolla 2012; Kasinitz et al. 2008).

Asylum seekers from Sudan entered Israel through the border with Egypt. Some were Muslims guided by Islamic Sharia law, and others were Christians who spoke Arabic and tribal languages. This community is characterized by significant heterogeneity regarding culture, religion, and language characteristics. In addition to religious beliefs, there are traditional beliefs among the various tribes. The Sudanese community in Israel is characterized by significant social cohesion and solidarity. There is a mutual guarantee and a sense of shared destiny. This is reflected in the internal establishment of community centers and aid initiatives for Sudanese refugees outside Israel.

The community of asylum seekers from Eritrea is very homogeneous regarding religion and culture; most of them are Orthodox Christians who mainly observe the traditional and ceremonial side of the church, such as marriage ceremonies, baptism, and mourning ceremonies. On the other hand, the Eritrean community is very heterogeneous regarding social characteristics; there are considerable differences between city people and rural people, between people with different levels of education, between those who speak English and those who do not, and between religious and secular people. The social structure is traditional and communal, centering on religion and the extended family (Shani 2015).

Israel is a Jewish, democratic, and liberal state. One of the self-evident characteristics of the liberal worldview is inter-religious tolerance. According to this view, the tolerant trend in Orthodox Judaism towards Christianity should have strengthened with the strengthening of liberalism. However, in the eyes of many of our contemporaries in Israel, Christianity is still seen as foreign worship. The Jewish hostility towards Christianity is not a thing of the past, but is still present and felt. Instead of patiently gazing toward the present and the future, Orthodoxy chooses to stay in the past and cling to its old suspicion of Christianity and its believers (HaCohen 2004). Furthermore, Ben Yohanan (2017) claims that religious intolerance and taking a strong stance towards other religions are apparently due to the changing political and cultural situation of the Jews. Only a few support inter-religious dialogue and hold moderate positions.

According to church officials, there has been a sharp increase in the number of attacks, hate crimes, and acts of vandalism against Christians in Jerusalem; people dare to spit on Christian clerics openly, and the police do not take the situation seriously enough and refuse to see the accumulation of incidents as a phenomenon (Hasson 2023).

### 1.1. Educational Aspect

Israel is a Jewish national state, without separation between religion and nationality. In Israel, secular Jews, liberal Jews, traditional Jews, religious Zionists, and ultra-Orthodox Jews live next door to one another. Each group has a different affinity to the issue of religion and state. At the same time, the State of Israel is characterized by multiple religious identities of Israeli communities living throughout the country, such as Bedouins, Druze, Circassians, and Arabs. In recent years, African Christian or Muslim asylum seekers have also arrived. This combination creates social and political tensions regarding the characteristics of the Jewish Israeli religious identity of the state. Today, the country maintains a "status quo" reflected in the education system, where each religious stream has separate educational institutions and adapted study programs (Sheleg 2013).

The Israeli government has determined that following the compulsory education law, children from families of asylum seekers and refugees without status staying in Israel will study in the public education system (Kemp and Raijman 2008; Green 2014). Studies that examined the communication methods of educators from the dominant majority group with parents from an ethnic minority group (Eliyahu-Levi and Ganz-Meishar 2019; Bashir et al. 2016; Deardorff 2009) found that teachers with intercultural competence that includes knowledge, skills, and openness towards different cultural settings create better personal relationships with parents. Therefore, the challenge facing educators in general, and Bible teachers in particular, is to mediate the perception of the Bible as a sacred text to the community of African Christian or Muslim asylum seekers while respecting their worldview, beliefs, and culture (Frugal and Gam-HaCohen 2021; Shemer 2016; Yona 2005).

### 1.2. Biblical Teaching

In the state education law, it is established that one of its main goals is "to teach the Torah of Israel, the history of the Jewish people, the heritage of Israel and the Jewish tradition, to instill the memory of the Holocaust and heroism, and to educate to respect them [ . . . ]". These principles reflect the humanistic, universal values of tolerance and love of humanity.

The question is whether teaching the Bible in elementary school preserves these universal values or pushes them away and promotes Jewish, religious, and national aspects over an emphasis on Hebrew cultural particularity. The answer is found on the supervisor's website of the Bible subject, where it is written: "The Bible is the foundational book of the people of Israel—the word of God to his people. The Bible tells the story of exemplary figures of the Jewish people for thousands of years. Its contents shape the life of the nation—its beliefs, aspirations, hopes, and leadership [ . . . ] from generation to generation, and instill knowledge, values, identification, and love for the book among the students".

The curriculum for teaching the Bible seems to reflect a religious worldview that prioritizes the Jewish, religious, and national aspects. Such an approach may have difficulty dealing with secular worldviews, with learners experiencing feelings of alienation, to bridge sociocultural gaps and promote universal values, freedom of thought, and cultural openness that illuminates the face of diverse communities.

The connection between Bible teaching and value education was discussed by researchers in the 20th century (Adar 1967; Greenberg 1979) and continues today (Shapira 2005; Amit 2010). The main discussion revolves around the question of what contributed to the decline of the value of the Bible in Israeli society in general and in state education in particular. Dor and De-Malach (2008) state that Bible teachers systematically avoid dealing with questions of values and morals and focus the responsibility for this on the Bible study programs and textbooks, which avoid questions of values and morals, thus causing teachers to miss an educational opportunity for discussing values in the classroom.

The considerable importance attributed to studying the biblical text and shaping a concept that singles out the sacred texts as a tool to connect with the divine being is closely related to the culture, calendar, rituals, symbols, and daily practice of Judaism, for example, the Simchat Torah holiday, the Bar Mitzvah celebration, the tallit in the canopy, setting a mezuzah at the door, and reading the Passover Haggadah (Guzman-Carmeli 2020).

### 1.3. Ideological and Pedagogical Identity

The teachers' knowledge, beliefs, and ideologies are related to personal and unique worldviews. Elbaz (1981) claims that teachers' self-concept is not always visible and is embedded in their pedagogical and ideological beliefs. According to Nisssan (1996), these perceptions reflect a person's identity, including elements adopted following past experiences and in plans for the future. Moreover, an individual's identity consists of values and loyalties that they are committed to and seek to preserve and give expression to. Hence, the concept of the professional–educational identity of teachers includes personal and professional components.

Beijaard et al. (2004) claim that professional identity is part of the "self", which includes beliefs, attitudes, life history, and personal narrative. According to Rodgers and Scott (2008), an educator's identity, in general, consists of four layers: (a) multiple historical, political, social, and cultural contexts; (b) mutual relations; (c) changes according to contexts and circumstances; and (d) stories that shape it. In the process of designing their professional identity, educators refine their personal and professional individuality and emphasize the unique qualities of their colleagues while examining the qualities similar to the groups to which they belong, such as family, friends, co-workers, nationality, and religion (Gee 2000–2001; O'Connor 2006).

Education researchers (Hishrik and Kfir 2012; Tabac and Hamu 2017; Antonek et al. 1997; Roberts 2000) also emphasized the use of reflective skills and reference to various interactions with colleagues, parents, experts, and students in the process of designing their professional identity. These skills mainly include inter-subjective aspects as manifested in the reflective process, in thinking about diverse perspectives, in confrontation with opposing perceptions, in a continuous process of interpretation, and in revealing an authentic personal voice.

In this study, we refer to the professional identity of Bible teachers as it emerges in a professional, cultural, religious, and national context with the desire to shape a Jewish Israeli identity and consciousness among students from refugee families and African asylum seekers in Israel.

This study aims to examine the ideological and pedagogical concepts, challenges, and actions of teachers in the field of biblical knowledge in multicultural elementary schools in which non-Jewish students are integrated.

## 2. Methods

This qualitative–interpretive study examines how Bible teachers face conflicts in teaching a biblical Jewish text in classes where non-Jewish African asylum-seeking students study (Zur and Eisikovits 2015; Zabar-Ben Yehoshua 2019). The study examines the multicultural and multilingual learning environment. It allows us to voice educators' personal and authentic voices as a first source on the teaching–learning processes of a religious text in a nationally, racially, religiously, culturally, and linguistically diverse classroom (Zur and Eisikovits 2015). In this way, we will better understand the struggles of those trying to provide an educational solution (Krumer-Nevo and Sidi 2012).

The research was conducted in collaboration with four elementary schools in the country's center where students from diverse groups differ from each other socioculturally, linguistically, religiously, nationally, and more. The children are veteran Israelis, Israeli Arabs, Jewish new migrants, Asian migrant workers, and African asylum seekers. The educational staff reveal a vision that advocates ensuring the right to and opportunity for quality and equal education for each student. The educational setting is a central anchor in the students' lives, giving them opportunities to develop their potential, acquire an education, and cultivate the life skills required for successful integration into Israel.

From Israeli society's point of view, as a whole, they are religious and secular Jews who believe in God; few Israelis extend to the atheist worldview, and there is almost no ideological secularism in Israel. Despite the tension between religion and state in Israel, the Jewish tradition strongly symbolizes Israeli personal identity (Sagi and Stern 2011).

The research participants are 12 Jewish Bible teachers: 11 women and 1 man. The teachers are between the ages of 32 and 48 years old with 5–25 years of experience teaching in state schools (and non-state religious Orthodox schools), and their teaching follows the curriculum of the Ministry of Education. As part of the professional development process, all teachers regularly participate in professional training in the field of knowledge, namely, teaching the Bible.

Four teachers are Orthodox religious, wear headscarves, and dress according to the rules of dress accepted in the ultra-Orthodox world. For example, they do not wear pants or a short-sleeved shirt, but cover the body completely. The remaining teachers lead a secular lifestyle while maintaining the Jewish tradition, especially on holidays. The teachers were chosen based on the researchers' professional or personal acquaintance with them due to their rich experience in teaching among diverse populations.

The research data were collected through personal in-depth interviews with each participant. The research tool is an in-depth interview through which it is possible to understand the experiences of the teachers and the meaning they attribute to their experiences. The research participants were interviewed in an in-depth interview based on guiding questions with minimal intervention by the researcher, since in this type of interview, there is a degree of flexibility and freedom of action for the interviewer and the interviewee, which may contribute to the research (Shkedi 2003). In addition, an in-depth interview makes it possible to obtain information on specific topics on the one hand; on the other hand, it makes it possible to understand human complexity without the limitation of a previous catalog (Fontana and Frey 2005).

The interviews took place with each participant in an educational setting and lasted for approximately 45 min. The interview began with a presentation of the purpose of the study, and all of the interviewees signed an informed consent form indicating their willingness to participate. In the next step, the interviewees answered guiding questions regarding the issues of teaching biblical knowledge in elementary school among Christian or Muslim African asylum-seeking students. Some examples of the questions include: What challenges do you face when teaching the Bible? How do you overcome these challenges? Please describe a case in a Bible lesson where you felt conflict or tension during the class. Are the parents involved in teaching–learning processes at the school? Describe the actions you perform in the process of planning the lesson and during the actual implementation. Why did you choose these actions?

After rereading the interviews and comparing them, the central themes were determined, which were repeated in the interviews (Lieblich et al. 2010). The findings were analyzed using qualitative content analysis, which includes dividing the information into contents and reorganizing them according to categories, allowing a fresh look at the research topics and revealing meanings that lie within the data. According to Shkedi (2003), content analysis is a window that enables the researcher to view the inner experiences of the participants and, in this study, focusing on the words and descriptions of the Bible teachers that reflect their perceptions, beliefs, attitudes, and pedagogical and organizational actions.

In the research, the ethical rules were carefully observed in order to maintain the anonymity and confidentiality of the respondents and the data, avoid offensive questions, and give the respondents a choice of whether to answer.

### 3. Findings and Discussion

In examining the ideological and pedagogical concepts, challenges, and actions of teachers in the field of biblical knowledge in multicultural elementary schools in which non-Jewish students are integrated, two content categories were found that reflect two tensions: (1) emotional religious tension and (2) pedagogical tension.

The findings indicate that Bible teachers are social and religious mediators in Israeli society. In the context of religious tension, teachers find themselves in situations of uncertainty, without the skills to help them bridge the gaps and soften the tensions; as a result, they are passive and remain silent. On the other hand, in terms of pedagogical tension, the teachers try to take the initiative, go beyond the boundaries of the familiar, and adapt the classroom activity to the culture of the country of origin and the everyday social contexts.

### 3.1. Emotional Religious Tension

The religious tension originates from the Jewish teacher's relatively empty multi-religious toolbox. Therefore, when teaching a sacred biblical text among Christian or Muslim children, they cannot provide an appropriate personal or classroom response. The teacher compares the teaching of the Bible in two different educational settings: the first teaching the Bible in a religiously homogeneous classroom, where Jewish children grew up on Bible stories and Bible heroes in their social contexts; and the second within a religiously heterogeneous class, where children from diverse socio-religious backgrounds study. In such an environment, one teacher testifies that she should be sensitive to the students' experiences. In her report, she refers to Sukkot, which is celebrated in September and is a biblical holiday. The Jews sit in the Sukkah (temporary structure) because the Israelites came out of Egypt and lived in the desert; they lived in the Sukkah because "[ . . . ] I made the people of Israel sit in Sukkot" (Leviticus 23:33):

> *The children see Sukkots in the public space on the balconies, in the yards, and at the entrance to the buildings. It is like a tent built outside without the context of the Israelites' exodus from Egypt. The children don't have the feeling of a holiday; there is no family experience of building a sukkah and decorating the sukkah. That is why I don't feel comfortable telling the biblical story or playing a familiar children's song related to Sukkot. I am sure if they had a choice they would ask what you want from us?! It is tough for me to give up my essence as a biblical teacher and the teaching of poetry, which can help in understanding the biblical story, but I have no choice.* (R.)

R. testifies to emotional and pedagogical difficulty. She cannot find an appropriate teaching method for the children from Christian and Muslim asylum-seeker families. She removes the content and does not teach it to the children. We can interpret her renunciation as pedagogical flexibility, openness, and understanding. In practice, relinquishing the biblical text, the songs, and the exposure to the tradition may harm their social integration. It leaves them outside as strangers, not belonging, which could intensify the difficulties and barriers they face. They are well aware of the power relations and the social and religious superiority of the white, Jewish Israeli man, who belongs to the dominant majority group. The teacher's decision not to expose them to the national, religious content in the public

space may increase the students' lack of belonging to the Israeli Jewish existence, the differentiation, the feeling of not belonging, and social rejection (Kasinitz et al. 2008).

Religious tension undermines the teacher's pedagogical concepts when the parents express their opinion and intervene in the conduct of the classroom on religious issues. An incident reported by D. took place on the Hanukkah holiday, a Jewish holiday celebrated for eight days to commemorate the victory of the Jews in the rebellion against the Greeks. The symbols of the holiday are a menorah, donuts, and spinning tops:

> *One of the Christian mothers loudly objected to bringing donuts to class—a Jewish custom. My first reaction was to oppose this mother because I wanted her to respect Jewish tradition, so I continued to bring the donuts to the children. But I was not at peace with myself, and on the way home, I continued to think, and I still didn't have an answer as to right and wrong.* (D.)

The fact that D. was not satisfied with the decision to bring donuts openly indicates that she does not know the correct pedagogical path she should have taken, indicating that D. holds pluralistic views and believes there is room for diverse cultural expression. However, bringing donuts proves that, in practice, she has an assimilation requirement, according to which all children of all religions should be similar to the Israeli Jews. She did not consider the interests of the other groups and did not recognize the differences between cultures and religions. Hence, she does not allow the cultivation of more than one ideal Israeli identity (Teter 2004; Yona 2005). The perceptual gap forced the teacher to adopt a policy of cultural assimilation as the only alternative.

From a critical point of view, we can argue that Africans in Israel have no chance of social mobility because their identity and foreignness are evident in the color of their skin (Hobbs 2010). The example illustrates that not all schools in Israel adopt multicultural pedagogy and partially recognize the value of the different cultures in Israeli society. Moreover, it seems that a change is required that will challenge the hegemonic discourse that denies racial diversity and makes room mainly for Jews who have a shared religious and cultural history reflected in the Hanukkah holiday's customs. Zur (2008) proves that Israeli society discriminates based on race, and skin color has significance in determining the life course of the country's citizens. In such a reality, educators should hold open discussions accompanied by a theoretical reference to the terms of ethnicity, origin, skin color, and power relations; all of these will also be reflected in the cultivation of intercultural competence as a service that will enable the beginning of an anti-racist change.

In this report, Bible teacher S. claims that Bible classes are the most appropriate place to have an authentic discussion about the differences between religions and between religious texts, and yet this does not happen:

> *Religious tension exists in the classroom, but it is not visible. I have not heard a child say I am a Jew or a Christian. Bible lessons are the most suitable for holding a discourse on the differences between religions, between the sacred texts and the different religions, but I don't dare to do that; I don't know how to do that. I only know that the religious differences create tension and hatred between the children, and they are not to blame; they would like to integrate, and I would like to help them. If I'm not Jewish, I don't belong here; I'm not a part of, and maybe I'm hurting the Jews—the religious tensions create the feeling of hatred for the next generation.* (S.)

S. seems to have not developed religious intercultural competence (Deardorff 2009). He is not skilled in conducting multicultural or multi-religious dialogue in class. He is unaware of the behaviors, beliefs, values, and heritage of the Christian or Muslim African asylum seekers' country of origin. He is well aware of the religious tension that creates hatred between the students, but fails to mediate and bring his pedagogical concepts to light in the classroom.

S.'s words, "Bible lessons are the most suitable for holding a discourse . . . But I dare not do so", illustrate the gap between perceptions, abilities, and actions. On the one hand, S. reveals a concept according to which dialogue may reduce tension, establish a better

learning process, and possibly even influence the children's future; on the other hand, S. lacks the pedagogical ability to lead the students through social processes that enable social mobility and perhaps even break through intergenerational barriers. Therefore, the school reality will continue to be such that racial, religious, and cultural ties may deepen the tensions and gaps between the dominant majority group and minority groups.

From a critical point of view, this example may prove that the standard policy in the education system in Israel is that of the hegemonic culture of the dominant and controlling groups in society. The teacher cannot provide an adequate pedagogical response to Christian and Muslim students who study in state schools, especially as religious issues are at the center of the lesson. Furthermore, the teacher perceives the African students who are not Jewish as having a low status in the social hierarchy, an excluded and discriminated against group. Therefore, he expresses empathy: "They are not to blame; they would like to integrate, and I would like to help them." These seem to be ambivalent feelings that imply superiority and paternalism. Labeling a minority group as inferior, even if accompanied by positive emotions, reinforces the positive image of a group, the dominant majority, and perpetuates racial discrimination (Fiske 2018).

Moreover, this finding is in line with the research of Herzog et al. (2008, p. 74), who claims: "Racism in Israel is institutionalized and carried by the branches of government, such as the police, and at the same time it is popular and a broad one that manifests itself in hatred of the foreigner." The social and religious differentiation is reflected in the appearance of skin color, which creates a racial hierarchy and leads to discrimination. The upper dominant majority group is white Jews versus black-skinned Christian or Muslim migrant families from Africa.

*3.2. Pedagogical Tension*

The pedagogical tension in the Bible lesson reflects the religious tension in the heterogeneous community in which a Jewish majority and a weakened African Christian or Muslim minority live together. According to the mandatory education law, all of the neighborhood children—Jews, Christians, and Muslims—attend the state school, and the teacher is required to find pedagogical principles that respond to the religious diversity in the classroom.

Religious diversity creates tension between the dominant majority group with power and supremacy and the silenced and invisible minority group in the public sphere. This tension is a crucial variable that affects the teaching activities, the pedagogical judgments, and the exposure of value perspectives of morality, conscience, and faith. The Bible teachers find themselves in a conflict between what is required in the curriculum and the requirements of the Ministry of Education, such as reading an original text, using the language of the Bible, teaching content about Judaism, and adapting teaching that responds to religious, emotional, and cultural differences.

Teacher C. shares the challenges she faces in teaching the biblical text:

*I have no tools or suitable study materials for African children, foreigners, Christians, or Muslims. No textbooks can help me adapt the content to their age and basic level of knowledge. They must have the book of the Bible as a text in Bible lessons. It is challenging how I will teach them a textbook without pictures and meditation when they do not understand Hebrew well. I have to teach Bible because I also have Jewish children in the class. I must not harm any student. Therefore, I have no choice; I do several things simultaneously, and not everything is visible.* (C.)

The responsibility for teaching the Bible in the classroom rests on the shoulders of the teacher and personal initiative for mediation actions and creating original teaching materials that will make the biblical story accessible to all students. Her words, "I must not harm any student", illustrate the fact that there is a risk that students from any religious group may be harmed or exposed to content that is not in line with their faith from home.

If C. teaches following the guidelines of the Ministry of Education, non-Jewish children may feel that they do not belong and are excluded, different, and discriminated against; on the other hand, if she emphasizes universal motifs of the biblical story in class, the Jewish majority group may claim religious harm.

The teacher's teaching experiences show that some mediation actions are conducted secretly, alone, and quietly. Teacher C.'s decision not to share with others proves that the education system does not provide a pedagogical response to teaching in heterogeneous spaces, but prefers to ignore it or assume there is no problem. She feels lonely and sometimes acts against the dictates of her Jewish conscience. Her pedagogical choice reflects teaching methods with a close pluralistic view as a narrative story with a moral.

Studies in teaching the biblical text (Litmanovitz 2019; Tayob 2018) indicate that to overcome religious tensions, teachers should mediate knowledge and values to students while emphasizing the biblical story in children's lives. One of the ways to meditate is through films, visual means, clothes, and particular texts adapted into songs and plays—all of these may bring the biblical text closer to the children's world and represent different religions in a more tangible and accessible way.

Here is an example of the activity of teacher Y., a religious woman with a head covering. She wrote a song for children on her own initiative that is an original and creative way to teach a biblical text in a religiously diverse classroom:

*I am a religious teacher and cannot convey my faith to the children because they are different. On the one hand, I had to teach them the Ten Commandments because I wanted the values I believe in to be passed on to them; on the other hand, I needed to know how I do it in front of them as strangers. I consulted my rabbi, and he said this is my mission. I started every morning with a song, and that's how I taught them the rules of behavior in the classroom. I called the song "The song of the day—for prayer in the hearts of all of us." I focused on the values and was excited when the parents shared that the children sang at home.* (Y.)

The biblical text is loaded with emotional content that strengthens the Jewish religion over others, such as the love of God and the Jews being a chosen people. For the religious teacher Y., these are strong concepts and beliefs that the non-Jewish students do not share. Y. found herself in a predicament, consulted the rabbi, and formulated a pedagogical concept capable of containing, understanding, and acting to mediate understanding and tolerance. She focused on social values, took the religious content out of its context, and made it more relevant to children. She offered the biblical message to all of the children. It is possible that the blurring of the Jewish religious ethos helped to reduce the tension and mobilize non-Jewish parents to understand the importance of Bible lessons.

Diversity in teaching methods is valuable in a heterogeneous religion class. A. added that the change that took place in her teaching methods is based on the integration of social values in the teaching of the biblical text:

*I realized that I could not memorize biblical texts with the children. I can't use the recitation skill as I know it. Memorizing knowledge about the world's creation will not encourage them to learn. I realized I had to ask social and value questions and indirectly teach the religious value.* (A.)

A. shows a commitment to activities that focus on questions of values in discussing the events that have considerations of morality and tolerance to mobilize the children and to teach the Bible, but indirectly. This finding is similar to the recommendation that having meaningful discourse and value discussions allows teachers to bring ethics and socially complex issues closer to the children's daily lives and deal with the conflicts that the Bible presents to us. All of this is born out of an awareness of the difficulties and a desire to look at them from an up-to-date perspective and deal with them while maintaining a respectful and cherishing attitude towards our founding texts (Zakovich 1995; Dor and De-Malach 2008; Magidov 2015; Katz and Katzin 2020).

The biblical language is archaic and less spoken in the children's daily life. In Bible lessons, the essence is understanding the text and writing answers. The children who speak Hebrew as their mother tongue know the stories and remember keywords and essential concepts. Migrant children whose mother tongue is not Hebrew are excluded from this experience. The Bible's vocabulary is critical for producing the message and understanding the meanings of the biblical story. A word-for-word conversion causes the biblical text to lose its authenticity. The language has value, cultural, and pedagogical importance in Bible teaching. The teacher should know how to navigate between focusing on the word's interpretation creatively, finding a semantic alternative, and maintaining the authenticity of the biblical language. Teacher S. shared that:

> *The language is complex; the level is very high. There is a problem with the words for children who have not yet acquired the language well, and Hebrew for them is very basic. I know they don't understand everything. Jewish children can handle the biblical language because they hear it at home and in other social contexts and are not ashamed to ask questions. On the other hand, it is a foreign language for foreign children, and they prefer to shut up in the classroom. I don't force them to participate, but I try to reach them each. When I pronounced the word Avraham with emphasis as in English, R.'s eyes opened. The student who kept interrupting me in class. He began to listen when he realized that the Bible is a book for all three religions.* (S.)

S.'s teaching methods correspond to the principles of Restorative Justice Pedagogy, such as dialogue or circles of peace, that enable students to learn how to share and listen with peers, set boundaries for moral and open dialogue, and promote tolerance and constructive engagement with each other's perspectives (Parker and Bickmore 2021). Therefore, the role of the teacher is more significant because it is impossible to understand the story without these words.

The teacher does not force the children to participate in the lesson and shows sensitivity to their visible and hidden objections. She tries to avoid an unequivocal statement that the children should study the biblical text and respects the complex situation. At the same time, her appeal to the student's world of religious knowledge tries to create trust while assuring him that he and his religion, and his personal life outside the classroom, are seen, heard, and respected. It seems that teacher S. reveals the ability to identify and understand the conceptual difference and reformulate biblical knowledge. These actions may allow the personal core values of the non-Jewish student to be present in the classroom, and may help overcome emotional and social tensions and encourage listening and interest in the lessons (Ogilvie and Fuller 2016).

N. reveals the issue of a gap between what is found in the school routine and what is desired in her educational concept:

> *The children ask questions, and I help them understand how the Torah stories connect to the stories they know from home. Many questions arise about Jesus and Christians. First, I explain that Judaism is the basis of Christianity and Islam, Jesus was a Jew, etc. This thing is very surprising to the children. I do not read from the religious text, but I tell it as a story with a plot and an educational message of global values and sometimes watch movies. The Israeli parents are occasionally angry, but I can do nothing. I would like the children to bring the stories they hear from home so they feel comfortable; for example, the parents record a story from Christianity and play it in the classrooms. But I know there's no chance in the world, so I don't offer.* (N.)

It is implied that the teacher is working to connect the Bible stories in the world of the children from Christian migrant families to levels they may have heard and know from home and from visiting the church. Certain words, "I explain first of all that Judaism is the basis of Christianity and Islam", were not said out of paternalism and a desire to prove the superiority of the Jewish majority group that rules Israel over a minority, but to share the knowledge she was educated and grew up with. As an Israeli Jewish educator, the teacher emphasizes the role of Judaism vis-à-vis other religions and perhaps strengthens the affinity

for Judaism. Although the teacher knows the children's difficulties in understanding the biblical story and language, she does not judge, hurt, or address them with insults. She formulates other teaching methods to maintain meaningful learning in the classroom (Lou and Noels 2020).

## 4. Conclusions

This study describes the personal voices of Bible teachers who face religious and pedagogical tension in a heterogeneous classroom regarding national, ethnic, religious, social, linguistic, and cultural differences. The research reveals the thoughts, experiences, interpretations, and beliefs regarding teaching a religious text in educational settings that include non-Jewish children.

The research findings reveal complex conflicts that reflect the ongoing and tense friction between institutional, political, legal, and religious issues in Israeli society. It also seems that the teachers detach themselves a little from the socio-historical background that shaped them, and make a significant effort to be worthy of teaching in heterogeneous classes in which the diverse Israel of the present is reflected. These teachers treat the Torah as a text that helps them build social bridges between people from the three religions of Abraham: Judaism, Islam, and Christianity.

Moreover, the findings shed light on the mechanisms that motivate educational frameworks and educational staff to encourage or weaken a disadvantaged ethnic group, namely, African Christian or Muslim asylum seekers. It was found that the Bible teachers show an ambivalent attitude towards this population: on the one hand, teachers show pedagogical activism and try to be creative and mediators of disciplinary knowledge content (teaching the biblical text). Despite this, it seems that even the teachers who recognize the differences in the knowledge of religious content, lifestyles, and customs between children from Jewish families and children from non-Jewish families fail to manage to reduce the gaps or bring these worlds together in school and outside.

On the other hand, teachers show passivity and silence, and ignore certain aspects of the situation. They reveal perceptions of cultural assimilation without regard for the interests of the minority group.

It is possible that the motive for the teachers' reaction, expressed in ignoring, closing their eyes, and sticking to the routine of school life by the Ministry of Education's guidelines, also stems from the lack of communication with the families of the parents of the African asylum seekers, who prefer to preserve their religious views as part of their identity and belonging to their country of origin. This reaction may also affect the reluctance of the children in Bible classes to be included, participate, and learn.

Furthermore, the reasons underlying the ambivalent attitude towards this population may also be related to the unique characteristics of this population, especially their skin color and cultural origin. The attitude of Israelis towards the skin color of Africans has a decisive role in preserving the stereotype of "blacks", which leads to perceptions and beliefs about their character, traits, and abilities.

From the global aspect of a world characterized by migration processes fraught with anxieties about foreigners, expressions of violence, and feelings of alienation, educators are required to understand that their teaching activities and interactions with students take place in a family, cultural, religious, and national context, and are not neutral or accidental. Most educational treatment mechanisms seem to be neither accessible nor adapted to the asylum seeker community. Therefore, the research is relevant to a global reality where the phenomenon of immigration is on the public agenda.

This study may add an essential layer to the body of knowledge regarding the understanding of the factors that influence Bible teachers when teaching in a heterogeneous classroom, while responding to differences and dealing with religious, national, linguistic, and sociocultural tensions. This understanding may promote a process for developing a professional identity among teachers and establishing relationships with people from vari-

ous cultures, to accept and understand them. In this way, we can deal with the migration challenges and integrate African asylum seekers in Israel.

Furthermore, the research findings emphasize the importance of expanding the pluralistic educational dialogue. With training colleges, education students should directly meet parents from minority groups and maintain an authentic conversation with social–educational activists in their community. These meetings may expose the students to other cultural and social knowledge, help them develop an understanding and empathy for the complex reality of life in social–religious contexts, and help them consolidate skills for managing a cooperative discourse that avoids expressions of racism, exclusion, and power. These educators may succeed in creating a dialogic learning space that allows a diverse, honest, and open discourse about family tradition, heritage, history, and life experience without the fear of paternalism. Such a space could increase the parents' involvement and strengthen the children's sense of competence.

**Author Contributions:** Both researchers, D.E.-L. and M.G.-M. make an equal contribution to this article. All authors have read and agreed to the published version of the manuscript.

**Funding:** This research received no external funding.

**Institutional Review Board Statement:** The study was conducted in accordance with the Declaration of Helsinki and approved by the Ethics Committee of the Levinsky-Wingate Academic College. Approval No.: 202303120.

**Informed Consent Statement:** Informed consent was obtained from all subjects involved in the study.

**Data Availability Statement:** The data collected in this research is unavailable due to privacy or ethical restrictions.

**Conflicts of Interest:** The authors declare no conflict of interest.

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
