# Peer review of "Teaching a Biblical Text among African Christian and Muslim Asylum-Seeker Children in Israel"

_religions, doi:10.3390/rel14040537_

Round 1
Reviewer 1 Report
The article offers an interesting contribution to a specif aspect, the one of teaching biblical texts to students moving to Israel from African countries and professing Islam and Christianity as their religions.
The main loophole in the full article in the lack of mention in the title of the context (Israel) where the study is carried out. I strongly reccommend a revision of the title with the inclusion of the term Israel. In general, the article is clear and issues are presented and discussed in a consistent way. Not being the topic part of my specific expertiese I am not in the postion to evaluate if references could be richer than the ones presented by the authors. Still all references reported in the study appeared to be fitting to the topic. Though English language and style are file perhaps an accurate revision is needed in the first half of the study. There are several spelling corrections to be made and repetition to be avoided.
I reccommend for publication after the above mentioned minor revisions
Author Response
Dear Reviewer (1),
Thank you for your efforts and insightful comments regarding the revised manuscript titled - Teaching a biblical text among African Christian and Muslim asylum-seeker Children in Israel
Specifically -
- We re-read the entire article, revised the English language, and corrected editorial issues and the repetition of topics.
- We checked all references in the manuscript and corrected them accordingly.
- Per your recommendation, we revised the title, including the term Israel.
Sincerely,
The Authors
Reviewer 2 Report
The article highlights a relevant topic in a complex, conflictual and suprematist context: the situation in Israel is not improving at all and the ongoing short-circuit tension between institution, political, juridical and religious issues is showing the complexity of a society whose inner souls is reaching a concrete point of collision.
The same article is clear, well written and elaborated in straight relation to the qualitative search. The main cognitive outcomes are shareable as the difficulties of some of the interviewed teachers whose main problem, moreover, seem to emerge in relation to detached themselves, a little, from their formative background, as in trying in being adequate to teach in the society that Israel is becoming today.
The "rich" world is and will continue to play an "attractive" role in the following decades towards less "lucky" people. At the same time, the Torah, in particular referring to Islam and Christianity, is a real helpful revelation in building bridges towards Abrahamic religions.
Sukkots' festivity, for example, is easily teachable because emphasizes as Jews had been emigrants in many phases of their religious history, with the possibility to trace easy connections with the huge majority of asylum seekers independently if they are Muslims or Christians.
The Pentateuch is the basement of Gospels and the Qur'an, and as reported Jesus was Jews; so the only concrete difficulties that seem to emerge in the article is the "open-minded" cognitive approach and ideological preconception that some teachers live in relation to their personal background.
In other words is the problem of a country that moved in the last decades to an ideological-political posture in contrast to the new-Babylonian (as anti-Nationalist) tendency in which the world is going.
Author Response
Dear Reviewer (2),
Thank you for your efforts and insightful comments regarding the revised manuscript titled - Teaching a biblical text among African Christian and Muslim asylum-seeker Children in Israel
Specifically -
- We re-read the entire article, revised the English language, and corrected editorial issues and the repetition of topics.
- We checked all references in the manuscript and corrected them accordingly.
- From your essential comments, we wrote a new paragraph in the concluding chapter of the article. A paragraph is highlighted in yellow.
Sincerely,
The Authors
Reviewer 3 Report
The subject of this article is an important one with implications for the position of minority cultures and religions within the state of Israel and for religious education in classes with a diversity of religions. It also adds to literature on the perspectives of religious education teachers negotiating differences between their own religion, that of their students and the requirements of the curriculum.
I would suggest the following minor changes to improve the article.
CONTENT
Context
The focus of this article is on pupils of migrant and African background. Some mention should be made of the Christian and Muslim families who originate from this part of the world, even if just to draw a distinction between their circumstances and those of new arrivals. Religious diversity in Israel is certainly not a new or migrant phenomenon. In the conclusion in particular and elsewhere in the article a lot is made of the racial (colour) element in the discrimination experienced by these communities. It would be worth mentioning the existence of religious-based intolerance outside the skin colour distinction , too - for example recent reports in the Jerusalem press on insults and assaults directed at Christian worship.
Do you have evidence of skin colour having an impact on the participant teachers' perspectives and pedagogies?
Concepts
Brief mention is made of Intercultural Competences with reference to Deardorff. It would be useful to the article to make more of these, relating the teachers' responses more closely to some recognised list of ICCompetences - such as Deardorff, or maybe the Council of Europe's model.
There is a mention of 'restorative justice pedagogy' l 417. There needs to be a (brief) explanationcof what this is and how the teacher's pedagogy relates.
Under methodology it would be good to have a very brief description of the teacher participants - gender, age, years of teaching, religion.
WRITING
l43-5 'This section .... it should'. This paragraph reads like instruction to the author rather than part of an article. It needs altering.
L 189f 'The findings are ....' etc. The tense here needs attention. The shift to present and then into future tense gives the impression that the analysis of the data has not been completed by the time the article is written, but is still to do.
Overstating.
In several cases the writing and language has loverstated the case where there is not the empirical evidence to support the contentions made. The article needs checking through for unsubstantiated claims or suggestions and language that is too definite. Sometimes rephrasing would help using words such as 'could', 'may', 'suggests', 'implies'
Examples include:
l 21 'must',
l 86 'will'
l 236 'intensifying' - I suggest 'could intensify' as an alternative
Similarly, 'increases the lack of belonging' - do we know this or is it conjecture?
l270 'must' - I suggest 'such a reality suggests a need for ..'
l 298 'will deepen' - I suggest 'could/may deepen'
Last paragraph - frequent use of 'will' implies certainty rather than hope or anticipation.
See also l 472-3 'This reaction also affects ...' - do we have evidence of this? Is this the teachers' assumption? If so do you have quotes to show this?
And l 483 'most educational treatment mechanisms are neither accessible nor adapted to the asylum seeker community' - do we have evidence for this strong statement? Is there other literature that argues this?
Lack of clarity
Page 9 first para
The shift of pronouns from 'he' to 'she' is confusing in this paragraph. Would changing 'The teacher' l 423 to 'A teacher using these principles' make it clearer?
l 249 is 'my mother' intended here or should it be 'this mother'?
l 112 the mention of 'the mother' is a bit surprising here. What do you mean?
l 120 are you talking about educators in general here or the educators who participated in Rodger's and Scott's research?
Typos
The following have been noted. There may be others:
l 75-77 'Whether ...' Should it be 'The question is whether ...' ?
Page 2 second paragraph. There is a " missing to show where the quote from the website ends.
l98 ' for discussion values'. Should this be 'for discussing values'?
Where an initial followed by . is used for a teacher the next word is often capitalised eg l 203 Reveals, Lacks; l 375-6, These, Found. These need correcting and the article checking for other such instances.
I 333 'Teacher C. She shares' Should this be 'Teacher C. shares'?
l 445 'to shares' should be 'to share'
See also
l 389 and 401 'still' not needed
l 264, 265 replace 'And' with 'furthermore', 'moreover', 'in addition' or just remove without replacement.
Author Response
Dear Reviewer (3),
Thank you for your efforts and insightful comments regarding the revised manuscript titled - Teaching a biblical text among African Christian and Muslim asylum-seeker Children in Israel
All changes are marked in yellow.
Specifically -
- We re-read the entire article, revised the English language, and corrected editorial issues and the repetition of topics.
- We checked all references in the manuscript and corrected them accordingly.
- Per your recommendation, we added a paragraph describing the study participants.
- We added an explanation about Restorative Justice Pedagogy and how it is related to the actions of teacher S. in the classroom.
- In the introductory chapter, we added information about the religious intolerance towards Christianity in Israel as a Jewish, democratic, and liberal state.
Sincerely,
The Authors